# Modeling Cyclic Crack Propagation in Concrete Using the Scaled Boundary Finite Element Method Coupled with the Cumulative Damage-Plasticity Constitutive Law

**DOI:** 10.3390/ma16020863

**Published:** 2023-01-16

**Authors:** Omar Alrayes, Carsten Könke, Ean Tat Ooi, Khader M. Hamdia

**Affiliations:** 1Institute of Structural Mechanics, Bauhaus Weimar University, Marienstraße 15, 99423 Weimar, Germany; 2School of Science, Engineering and Information Technology, Federation University, Ballarat, VIC 3350, Australia; 3Institute of Continuum Mechanics, Leibniz Universität Hannover, 30167 Hannover, Germany

**Keywords:** crack propagation, cohesive zone method, constitutive modelling, cyclic loading, scaled boundary finite element

## Abstract

Many concrete structures, such as bridges and wind turbine towers, fail mostly due to the fatigue rapture and bending, where the cracks are initiated and propagate under cyclic loading. Modeling the fracture process zone (FPZ) is essential to understanding the cracking behavior of heterogeneous, quasi-brittle materials such as concrete under monotonic and cyclic actions. The paper aims to present a numerical modeling approach for simulating crack growth using a scaled boundary finite element model (SBFEM). The cohesive traction law is explored to model the stress field under monotonic and cyclic loading conditions. In doing so, a new constitutive law is applied within the cohesive response. The cyclic damage accumulation during loading and unloading is formulated within the thermodynamic framework of the constitutive concrete model. We consider two common problems of three-point bending of a single-edge-notched concrete beam subjected to different loading conditions to validate the developed method. The simulation results show good agreement with experimental test measurements from the literature. The presented analysis can provide a further understanding of crack growth and damage accumulation within the cohesive response, and the SBFEM makes it possible to identify the fracture behavior of cyclic crack propagation in concrete members.

## 1. Introduction

Concrete structural elements very often fail due to fatigue fractures, in which repeated loading can lead to the growth of existing cracks [1,2,3,4]. To better understand the fatigue fracturing under cyclic loading, a detailed analysis of the fatigue behavior and the associated crack propagation is required for economical and reliable design of concrete structures.

The advanced studies on cyclic crack propagation are mostly empirical, wherein large number of data samples from experiments are used for fitting the relationship. The most commonly used approach to predict fatigue life and crack growth rate is the well-known Paris law [5,6]. This phenomenological law relates the amplitude of the stress state (defined by stress intensity factor K) and the crack growth rate da/dN, which can be considered a valuable tool for engineering fatigue analysis. However, it has been shown that Paris law loses much of its prediction ability when conditions are not ideal, such as with non-constant amplitude loading and short cracks [7,8]. Nevertheless, advanced numerical models have been developed widely to capture the phenomena behind the cyclic crack propagation under subcritical loading levels. Numerical simulations are more flexible in the sense that they can predict fatigue life and crack growth under general loading conditions and geometries. They can be applied to study design variations in early design stages.

Several modeling approaches for crack propagation under cyclic and fatigue loading are well documented in the literature [9,10]. The cohesive zone model (CZM) has been implemented in classical fracture mechanics by [11,12] to reduce the mesh quality required for crack simulation. The CZM is based on elastic damage material for both monotonic and fatigue crack growth [13,14]. For concrete material, the softening damage, whose localization is governed numerically by finite element simulation, is aimed at simulating the propagation of the fatigue fracture in the cohesive process zone [15,16]. However, these types of models are used to accumulate damage only along the damaged locations of the loading/unloading paths.

The second type of crack simulation model is the phase field model (PFM). The concept of the PFM approach is to regularize free energy of degradation, which effectively reduces material fracture resistance under fatigue loading [17]. It was developed to predict quasi-static and dynamic fracturing in brittle and ductile regimes considering isotropic and anisotropic toughness [18]. This method introduces the degradation of the fracture energy as a function of a local energy-accumulation variable. As a result of repeated loading, the structural loading history is taken into consideration [19]. Similar approaches have been published recently in [20], which simulated fatigue crack growth. A nonlinear kinematic and isotropic hardening were considered. Differently, simulations of molecular dynamics can be used to evaluate the interfacial strength [21].

Additionally, discrete lattice models have many features of the discrete element method (DEM) to simulate the heterogeneous microstructure and crack propagation [22]. The formulation combines the damage mechanics and plasticity theory with a cyclic damage evolution law. The model characterizes the critical response of concrete material undergoing cyclic loading. The behavior obtained by the DEM simulations is a collective response constituted from all contacts and particles in the domain.

Many models in the literature [23,24,25,26,27,28,29,30,31] are dedicated to simulating the quasi-brittle behavior, including a set of constitutive equations for the monotonic, fatigue, and hysterical material responses. Furthermore, several calculation schemes also exist to predict tensile, flexural monotonic, and fatigue behavior [32,33]. The established damage law allows a damage accumulation process for random cycles. The damage model concludes the primary dissipative phenomenon, which is activated during unloading and reloading.

The scaled boundary finite element method (SBFEM) is a very attractive approach to modeling crack nucleation and propagation under general loading conditions [34,35,36,37]. The cohesive fracture and stress field can be determined using interface elements with zero thickness, which are inserted directly into the SBFEM [38,39,40]. The cohesive traction forces and the stress field close to the crack tip are accurately computed as they are defined analytically. This enables the onset of crack propagation to obtain the correct load-deflection response. Yang [41] developed the SBFEM to solve linear crack propagation in brittle materials under monotonic loading. He benefited from the salient feature of the high accuracy of the stress intensity factor (SIF) in SBFEM computed directly from singular stress solutions and flexible substructuring of each domain. The crack simulation of concrete slabs based on a cohesive zone model in an explicit SBFEM-FEM frame for seismic cyclic loading was reported in [42] to facilitate dynamic analysis. However, the calculation of coupled SBFEM-FEM analysis can be very computationally intensive. For cyclic loading, the crack evolution can also be simulated using quasi-static analysis. The accuracy of the method was validated by a cyclic damage test with a concrete beam. A fully automatic modeling methodology characterized by a simple remeshing algorithm was developed, and the mixed-mode crack propagation problem was efficiently solved. Yang and Deeks [43] further coupled the procedure of SBFEM with the FEM for quasi-brittle materials. An extended polygon scaled boundary finite element method [44] was developed to simulate nonlinear dynamic analysis. A direct remeshing algorithm for crack propagation was obtained for quasi-brittle materials. The study of dynamic fracture modeling by SBFEM was developed in [45] to model the crack propagation of impact-test specimens. The stress intensity factor, displacement, and stresses were extracted from the dynamic solution.

In the present paper, we further extend the SBFEM for modeling cyclic-damage-induced cracks’ behavior within the SBFEM framework. The model considers the cumulative crack opening/sliding measure to dominate the damage mechanism at the subcritical loading levels. Similar approaches have been proposed in [40] for the numerical simulation of concrete under monotonic loading. The novelty of our approach is to establish a link between the cyclic damage rate and the efficiency of the SBFEM in modeling crack propagation. By comparing the thermodynamic softening law of the constitutive model for fracture, several aspects have been provided, which include the loading–unloading path, the damage evolution during the load cycle, and the crack-opening traction behavior.

The paper is organized as follows. The theoretical formulation of the cohesive crack model inside SBFEM is represented in Section 2. The behavior of the constitutive material model is studied at the level of material point (Gauss point) in Section 3. The performance of the cohesive cyclic crack model within the thermodynamic framework is then reported, which was applied in [46]. In Section 4, we present the calibration and validation of the model based on the results of the cyclic flexural bending test of plain concrete published in the literature. We present numerical investigations focused on the effect of the loading sequence on the material behavior.

## 2. Scaled Boundary Finite Element (SBFEM)

### 2.1. Fundamentals

Figure 1 shows the basic concept of the cohesive crack model in the scaled boundary method for a typical bounded domain. The mesh is represented by a discretized collection of arbitrary-sided polygons, or (as in Figure 1a) quadtrees elements. Each element is maintained by a curve relative to a scaling center ($x_{0}$,$y_{0}$). This condition is satisfied by dividing the domain into many sub-domains, which can be made visible for each boundary. The boundary is discretized by one-dimensional finite elements with a local coordinate η in an interval of −1 ≤η≤ 1; see Figure 1b. Let ($x_{0}$,$y_{0}$) be the scaling center, and ξ is the radial coordinate with ξ = 0 at the center and ξ = 1 at the boundary. The coordinates on the boundary are interpolated by $x_{b}=\left[N\left(\eta \right)\right]\left\{x_{bn}\right\}$, and $y_{b}=\left[N\left(\eta \right)\right]\left\{y_{bn}\right\}$, where [N(η)] is the vector of nodal shape functions, and $\left\{x_{bn}\right\},\left\{y_{bn}\right\}$ are the nodal coordinates. The displacement field, u(ξ,η), can be defined semi-analytically as
(1)$$\left\{u(\xi ,\eta )\right\}=\left[N_{u}\left(\eta \right)\right]\left\{u\left(\xi \right)\right\}$$

We calculate the nodal displacement functions u(ξ) at the radial lines, ξ. Meanwhile, they are interpolated by the linear shape functions $\left[N_{u}\left(\eta \right)\right]$ in the direction of η, which are obtained by multiplying a suitable identity matrix with each element in [N]. Thus, the strain and the stress fields are formulated as: (2)$$\left\{\varepsilon \left(\xi ,\eta \right)\right\}=\left[B_{1}\left(\eta \right)\right]{\left\{u\left(\xi \right)\right\}}_{,\xi }+\left(1/\xi \right)\left[B_{2}\left(\eta \right)\right]\left\{u\left(\xi \right)\right\}$$
(3)$$\left\{\sigma \left(\xi ,\eta \right)\right\}=\left[D\right]\left\{\varepsilon \left(\xi ,\eta \right)\right\}=\left[D\right]\left[B_{1}\left(\eta \right)\right]{\left\{u\left(\xi \right)\right\}}_{,\xi }+\left(1/\xi \right)\left[B_{2}\left(\eta \right)\right]\left\{u\left(\xi \right)\right\}$$
where $B_{1}\left(\eta \right)$ and $B_{2}\left(\eta \right)$ are the strain matrices, and *D* is the constitutive matrix [39]. The weak form of the elastic equilibrium of forces is obtained according to the principle of virtual work [47], or from the weighted residual technique; see ref. [34]. The governing equations can be written as follows: (4)$$\left[E_{0}\right]\xi^{2}{\left\{u\left(\xi \right)\right\}}_{,\xi \xi }+\left(\left[E_{0}\right]+\left[E_{1}\right]+\left[E_{1}^{T}\right]\right)\xi {\left\{u\left(\xi \right)\right\}}_{,\xi }-\left[E_{2}\right]\left\{u\left(\xi \right)\right\}=0$$
(5)$$\left\{P\left(\xi \right)\right\}=\left[E_{0}\right]\xi {\left\{u\left(\xi \right)\right\}}_{,\xi }+{\left[E_{1}\right]}^{T}\left\{u\left(\xi \right)\right\}$$
with {P} being the load vector. Equation (Equation 4) includes second-order Cauchy–Euler equations, called the scaled boundary finite element equation in the displacement with the coefficient matrices $\left[E_{0}\right],\left[E_{1}\right],\left[E_{2}\right]$. Furthermore, Equation (Equation 4) is a homogeneous second-order differential Equation (in case there is no side face or body loads) with *n* unknowns. By introducing a new variable [χ(ξ)] with Hamiltonian matrix *Z*, the system becomes a first-order ordinary differential equation [48] as
(6)$$\xi {\left[\chi \left(\xi \right)\right]}_{,\xi }=-\left[Z\right]\left[\chi \left(\xi \right)\right]$$
and
(7)$$\left[\chi \left(\xi \right)\right]={\left[\left\{u\left(\xi \right)\right\}\left\{q\left(\xi \right)\right\}\right]}^{T}$$
where q(ξ) are analytical functions that represent the internal nodal forces vector: (8)$$\left\{q\left(\xi \right)\right\}=\left[E_{0}\right]\xi {\left\{u\left(\xi \right)\right\}}_{,\xi }+{\left[E_{1}\right]}^{T}\left\{u\left(\xi \right)\right\}$$
and the Hamitonian matrix is calculated as a function of $\left[E_{0}\right],\left[E_{1}\right],\left[E_{2}\right]$: (9)$$\left[Z\right]=\left[\begin{array}{cc} {\left[E_{0}\right]}^{-1}{\left[E_{0}\right]}^{T} & -{\left[E_{0}\right]}^{-1}\\ -\left[E_{2}\right]+\left[E_{1}\right]{\left[E_{0}\right]}^{-1}{\left[E_{1}\right]}^{T} & -\left[E_{1}\right]{\left[E_{0}\right]}^{-1} \end{array}\right]$$

An eigenvalue decomposition of [Z] follows [49]: (10)$$\left[Z\right]\left[\begin{array}{cc} \left[\phi_{u}^{\left(n\right)}\right] & \left[\phi_{u}^{\left(p\right)}\right]\\ \phi_{q}^{\left(n\right)}] & \left[\phi_{q}^{\left(p\right)}\right] \end{array}\right]=\left[\begin{array}{cc} \left[\phi_{u}^{\left(n\right)}\right] & \left[\phi_{u}^{\left(p\right)}\right]\\ \left[\phi_{q}^{\left(n\right)}\right] & \left[\phi_{q}^{\left(p\right)}\right] \end{array}\right]\times \left[\begin{array}{cc} \left[\lambda^{\left(n\right)}\right] & 0\\ 0 & \left[\lambda^{\left(p\right)}\right] \end{array}\right]$$
where [λ] is the diagonal matrix of $\lambda^{\left(p\right)}$ and $\lambda^{\left(n\right)}$. The superscripts p and n refer to positive and negative. $\left[\phi_{q}^{\left(p\right)}\right]$, $\left[\phi_{u}^{\left(p\right)}\right]$, and $\left[\phi_{u}^{\left(n\right)}\right]$ are the eigenvectors corresponding to $\lambda^{\left(p\right)}$, $\left[\phi_{q}^{\left(n\right)}\right]$, and $\left[\lambda^{\left(n\right)}\right]$, respectively. The solution of Equation (Equation 6) yields: (11)$$\left\{q\left(\xi \right)\right\}=\left[\phi_{q}^{\left(n\right)}\right]\xi^{-[\lambda^{\left(n\right)}]}\left\{c^{\left(n\right)}\right\}+\left[\phi_{q}^{\left(p\right)}\right]\xi^{-[\lambda^{\left(p\right)}]}\left\{c^{\left(p\right)}\right\}$$
(12)$$\left\{u\left(\xi \right)\right\}=\left[\phi_{u}^{\left(n\right)}\right]\xi^{-[\lambda^{\left(n\right)}]}\left\{c^{\left(n\right)}\right\}+\left[\phi_{u}^{\left(p\right)}\right]\xi^{-[\lambda^{\left(p\right)}]}\left\{c^{\left(p\right)}\right\}$$$\left\{c^{\left(p\right)}\right\}$ and $\left\{c^{\left(n\right)}\right\}$ are the integration constants. For a bounded domain, the boundary condition at {ξ=0} produces $\{c^{\left(p\right)}\}=0$. In this case, the modes of non-positive real components of eigenvalue [λ] contribute to the solution of finite displacement at the scaling center.

The equivalent nodal forces on the boundary and the stiffness matrix of the domain are formulated, respectively, as
(13)$$\left\{P\right\}=\left[\phi_{q}^{\left(n\right)}\right]\left\{c^{\left(n\right)}\right\}=\left[\phi_{q}^{\left(n\right)}\right]{\left[\phi_{u}^{\left(n\right)}\right]}^{-1}\left\{u_{b}\right\}$$
(14)$$\left[K\right]=\left[\phi_{q}^{\left(n\right)}\right]{\left[\phi_{u}^{\left(n\right)}\right]}^{-1}$$

At the boundaries, the nodal displacements $\left\{u_{b}\right\}$ can be calculated from the global stiffness matrix *K* and load vector *P*.

Meanwhile, substituting Equation (Equation 11) into Equation (Equation 1) yields the displacement field in the bulk domain as
(15)$$\left\{u(\xi ,\eta )\right\}=\left[N_{u}\left(\eta \right)\right]\sum_{i=1}^{n}\xi^{-\lambda^{{\left(n\right)}_{i}}}c_{i}\left\{\phi_{i}\right\}$$

Hence, the stress field is formulated by
(16)$$\left\{\sigma \left(\xi ,\eta \right)\right\}=\left[D\right]\sum_{i=1}^{n}\xi^{-\lambda^{{\left(n\right)}_{i}}}\left(\left[-\lambda^{\left(n\right)}\right]\left[B_{1}\left(\eta \right)\right]+\left[B_{2}\left(\eta \right)\right]\right)\left\{\phi_{i}\right\}$$

### 2.2. Stress Field at Crack Tip with Cohesive Tractions

The fracture process zone in a quasi-brittle material can transfer the cohesive forces between the crack faces. This is attributed to the interlocking of the aggregate, in addition to the surface friction. The cohesive traction representing the crack faces is applied as side-face forces. The equilibrium condition (Equation (Equation 4)) in a polygon containing a crack tip is augmented to include the load vector containing the side-face tractions, as in [43].
(17)$$\left[E_{0}\right]\xi^{2}{\left\{u\left(\xi \right)\right\}}_{,\xi \xi }+\left(\left[E_{0}\right]+\left[E_{1}\right]+\left[E_{1}^{T}\right]\right)\xi {\left\{u\left(\xi \right)\right\}}_{,\xi }-\left[E_{2}\right]\left\{u\left(\xi \right)\right\}-\left\{F_{t}\left(\xi \right)\right\}=0$$

In this work, the cohesive force on the crack faces $\left\{F_{t}\left(\xi \right)\right\}$ will be computed based on the shadow domain procedure, which has been introduced by [40].

The concept of the cohesive cyclic crack model, as depicted in Figure 1, is shown in the following steps:The mesh generation of the domain in Figure 1a and the cohesive zone in the surroundings of the crack polygon is defined. In this method, the generic mesh contains an arbitrarily many sided polygon in boundary regions, master cells far away from the boundaries, and the crack cells.The crack cell is divided into two SBFEM cells to discretize the crack faces and to insert the interface elements into the SBFEM system. The local coordinates ξ,η of the SBFEM system are illustrated in Figure 1b.The shadow domain is generated as shown in Figure 1c. It is implemented in order to calculate the cohesive tractions (side-face forces) and the nodal displacements throughout the crack subdomain. This method inserts a node at the crack tip with three corresponding edges (two edges, $L_{1}$ and $L_{2}$, for each crack face, and one edge, $L_{3}$, to split the crack cell into two). Knowing the crack angle, θ, the orientation of $L_{3}$ is projected in a way that a straight line is extended from the crack tip with an angle θ. Then, the node closest to the intersection point at edge of the cracked cell is employed to split the polygon.The SBFEM is directly coupled with zero-thickness, four node-interface elements along the crack path (Figure 1d) which are inserted along the lines of the mesh. The cohesive edges $\left(N_{1},\;\;N_{2},\;\;N_{3},\;\;N_{4}\right)$ divide the subdomains into two divisions. The pair $\left(N_{1,u}\;\;N_{3,u}\right)$ and $\left(N_{2,u}\;\;N_{4,u}\right)$ form contact pairs with a set of crack opening (w). Additionally, the pair $\left(N_{1,v}\;\;N_{3,v}\right)$ and $\left(N_{2,v}\;\;N_{4,v}\right)$ form contact pairs with a set of crack sliding (s). As the crack propagates, the interface element domain is inserted into the mesh. This can satisfy the compatibility condition in the displacement between the SBFEM polygons and the interface elements.Along the crack paths, the fracture process zone is characterized using softening laws of the thermodynamics; see Figure 1e. For concrete, the softening behavior for crack opening and sliding proposed model is based on [46] and defined in the next section. The model uses the cumulative measure of slip as a fundamental damage driving mechanism at the subcritical levels of loading.

In the fracture process zone, cohesive tractions $t_{n},t_{s}$ are expressed as a function of relative opening and sliding displacements *d*. In the local coordinate system, the stiffness matrix reads: (18)$$\left[k_{int}\right]=\frac{A}{2}\sum_{i=1}^{n_{g}}w_{i}{M_{i}}^{T}\left[k\right]M_{i}$$
where *k* is the stiffness of the softening laws, *A* is the crack surface area, $w_{i}$ is the one-dimensional Gaussian weight, $n_{g}$ is the number of integration points, and $M_{i}$ is the linear shape function matrix [40]. The stiffness matrices of the interface element $k_{int}$ can be assembled attractively. In this case, the local coordinates ($\xi_{p}$, $\eta_{p}$) in the shadow domain are defined first to obtain the coordinates (x,y) for a new node in the new crack cell. For this purpose, we use a search algorithm to determine the element in the shadow domain that includes the point (x,y). In doing so, the nodal displacements and the cohesive tractions are calculated along the crack. These are then mapped back to the crack cell to calculate the stress intensity factors required to determine if the crack propagates. The SIF considering the cohesive forces on the crack face is calculated by representing the cohesive forces as a power function in ξ following from the form of the side face traction vector $F_{t}\left(\xi \right)$, as in [43].

Linearly varying or constant distributed loads are approaches to representing a force over a particular distance. According to [47], when the side-face loads are distributed by a power function, then the modal displacement loads are
(19)$$\left\{u_{t}\left(\xi \right)\right\}=\xi^{t+1}\left\{\phi_{t}\right\}$$

Substitution of Equation (Equation 19) into Equation (Equation 17) yields
(20)$${\left[{\left(t+1\right)}^{2}\left[E_{0}\right]+\left(t+1\right)\left(\left[{E_{1}}^{T}\right]-\left[E_{1}\right]\right)-\left[E_{2}\right]\right]}^{-1}\left\{\phi_{t}\right\}+\left\{F_{t}\right\}=\left\{0\right\}$$

Rearranging will give the nodal displacements for the side-face load mode $\left\{\phi_{t}\right\}$ as
(21)$$\left\{\phi_{t}\right\}=-{\left[{\left(t+1\right)}^{2}\left[E_{0}\right]+\left(t+1\right)\left(\left[{E_{1}}^{T}\right]-\left[E_{1}\right]\right)-\left[E_{2}\right]\right]}^{-1}\left\{F_{t}\right\}=\left[B_{1}\left(t\right)\right]\left\{F_{t}\right\}$$

In order to express the cohesive tractions as a power of function, the normal traction distribution σ(ξ) is assumed to be the summation of *M* raised to the power of function ξ:(22)$$\sigma \left(\xi \right)=f_{t}\sum_{i=1}^{M}e_{i}\xi^{ti}$$
where $e_{i}$ is coefficient to be calculated. Considering a parameter μ, the exponents $t_{i}$ are determined as $t_{i}=\left(i-1+\mu \right)$.

The tractions at the crack tip, the Gaussian points, and the crack mouth $\sigma_{j}$(*j* = 1, *M*) are used to generate *M* number of equations as
(23)$$\sigma_{j}=\sigma \left(\xi_{j}\right)=f_{t}\sum_{i=1}^{M}e_{i}\xi_{j}^{ti}$$
where $\xi_{j}=l_{j}/L$ is the distance from the jth point on the crack to the crack tip $l_{j}$ and the length of the crack *L*. The coefficients $\left\{e\right\}={\left\{e_{1}\;\;e_{2}\;...\;e_{M}\right\}}^{T}$ are then calculated as
(24)$$\left\{e\right\}={\left[S\right]}^{T}{f_{t}}^{-1}\left\{\sigma \right\}$$
where $\left\{\sigma \right\}={\left\{\sigma_{1}\;\;\sigma_{2}\;...\;\sigma_{M}\right\}}^{T}$, and the matrix [S] is
(25)$$\left[S\right]=\left[\begin{array}{cccc} \xi_{1}^{t_{1}} & \xi_{1}^{t_{2}} & \cdots & \xi_{1}^{t_{M}}\\ \xi_{2}^{t_{1}} & \xi_{2}^{t_{2}} & \cdots & \xi_{2}^{t_{M}}\\ \vdots & \vdots & \ddots & \vdots \\ \xi_{M}^{t_{1}} & \xi_{M}^{t_{2}} & \cdots & \xi_{M}^{t_{M}} \end{array}\right]$$

The nodal side-face load vector becomes
(26)$$\left\{F_{t}\left(\xi \right)\right\}=\sum_{i=1}^{M}\xi^{t_{i}}\left\{F_{t_{i}}\right\}$$
with
(27)$$\left\{F_{t}\left(\xi \right)\right\}=Af_{t}e_{i}\left\{R_{1}\right\}$$
and
(28)$$\left\{R_{1}\right\}={\left\{-sin\delta \;\;cos\delta \;\;0\;\;\cdots \;\;0\;\;sin\delta \;\;-cos\delta \right\}}^{T}$$
where A= is the area of crack surface.

The displacement solution is thus expressed by two components: the modes of normal displacement due to external loading and the modes of side-face displacement due to cohesive tractions as
(29)$$\left\{u(\xi ,\eta )\right\}=\left[N\left(\eta \right)\right]\left[\sum_{i=1}^{N}c_{i}\xi^{\lambda_{i}}\left\{\phi_{i}\right\}+\sum_{i=1}^{M}e_{i}\xi^{t_{i}+1}Af_{t}\left[B_{1}\left(t_{i}\right)\right]\left\{R_{1}\right\}\right]$$

On the subdomain boundary, the nodal displacement $u_{bs}$ is calculated as
(30)$$\left\{u_{bs}\right\}=\left[\phi \right]\left\{c\right\}+\left[\phi_{t}\right]\left\{e\right\}$$
where [ϕ] and $\left[\phi_{i}\right]$ are given in Equation (Equation 10), and the matrix $\left[\phi_{t}\right]$ is transformed as: (31)$$\left\{\phi_{t}\right\}=Af_{t}\left[B_{1}\left(t_{1}\right)\;\;B_{1}\left(t_{2}\right)\;\;\cdots \;\;B_{1}\left(t_{M}\right)\right]\left\{R_{1}\right\}$$

The nodal displacements $u_{bs}$ in Figure 2 are gained by mapping the mesh from the shadow domain, as shown in Figure 2b. The constants {c} are given by
(32)$$\left\{c\right\}={\left[\phi \right]}^{-1}\left(\left\{u_{bs}\right\}-\left[\phi_{t}\right]\left\{e\right\}\right)$$

Subsequently, Equation (Equation 29) is read as: (33)$$\left\{u(\xi ,\eta )\right\}=\left[N\left(\eta \right)\right]\sum_{i=1}^{N+M}c_{i}\xi^{\left(\bar{\lambda_{i}}-1\right)}\left\{\bar{\phi_{i}}\right\}$$
where
$$\left\{\begin{array}{cc} \bar{\phi_{i}}=\phi_{i},\;\bar{\lambda_{i}}=\lambda_{i}\; & \mathrm{for}\;i=1,\cdots ,N\\ \bar{\phi_{i}}=\phi_{t},\;\bar{\lambda_{i}}=t_{i}+1\; & \mathrm{for}\;i=N+1\cdots ,N+M \end{array}\right.$$

The stress field can be calculated similarly to Equation (Equation 16) as
(34)$$\left\{\sigma (\xi ,\eta )\right\}=\sum_{i=1}^{N+M}c_{i}\xi^{\left(\bar{\lambda_{i}}-1\right)}\left\{\psi_{i}\left(\eta \right)\right\}$$
where each term in Equation (Equation 34) can be interpreted as a stress mode and
(35)$$\left\{\psi_{i}\left(\eta \right)\right\}=\left[D\right]\left(\bar{\lambda_{i}}\left[B_{1}\left(\eta \right)\right]+\left[B_{2}\left(\eta \right)\right]\right)\left\{\bar{\phi_{i}}\right\}$$

Comparing Equation (Equation 15) and Equation (Equation 33), and Equation (Equation 16) and Equation (Equation 34) shows that when the cohesive traction is evaluated, an extra number (*M*) of displacement nodes and the same of stress modes are added to the displacement field and stress field, respectively.

The direction of crack propagation is then determined based on [43]. In order to consider a perfect crack path prediction, the SIFs of the semi-analytical SBFEM stress solutions are calculated.

### 2.3. Stress Intensity Factor (SIF) for Scaling Center at Crack Tip

The SBFEM has the advantage of accurately representing the crack zone’s stress field without needing a more discretized mesh [38,50]. This tool enables the SIFs to be directly calculated from the semi-analytical solutions of the stresses. In this work, two SIFs are determined. The first is obtained from the linear elastic fracture mechanics solution at a generic load step and is used to determine the crack propagation direction. The side-face tractions are not considered in this case. The second concerns the crack cell considering the effect of the cohesive tractions obtained from the shadow domain. In both cases, the procedure to calculate the SIFs is the same. The only difference is the equation used to represent the stress field, i.e., Equation (Equation 16) in case 1 and Equation (Equation 34) in case 2. The procedure is outlined as follows:

Figure 1c shows the cracked domain modeled by the SBFEM. The location of the scaling center should be at the crack tip. There is no need to discretize the side faces connected to the scaling center. The SIF could be accurately computed from the semi-analytical solutions of the stresses [51]. The stress intensity factors solutions can be extracted from their definitions as follows.
(36)$$\left\{\begin{array}{c} K_{I}\\ K_{II} \end{array}\right\}=\lim_{r\to 0}\left\{\begin{array}{c} \sqrt{2\pi r}\sigma_{yy}{|}_{\theta =0}\\ \sqrt{2\pi r}\sigma_{xy}{|}_{\theta =0} \end{array}\right\}$$
where *r* and θ represent the polar coordinates. As illustrated in Figure 1, *r* and θ originate at the crack tip and are correlated by
(37)r=ξL(θ)
where L(θ) is the distance between any point A at the cracked domain and the crack tip ($L\left(\theta \right)=L_{3}$ in Figure 1c). Substituting Equation (Equation 37) in Equation (Equation 36) leads to
(38)$$\left\{\begin{array}{c} K_{I}\\ K_{II} \end{array}\right\}=\lim_{r\to 0}\left\{\begin{array}{c} \sqrt{2\pi L(\theta )}\sum_{i=0}^{n}c_{i}\xi^{-\lambda i-1}\sigma_{yy}{|}_{\theta =0}\\ \sqrt{2\pi L(\theta )}\sum_{i=0}^{n}c_{i}\xi^{-\lambda i-1}\sigma_{xy}{|}_{\theta =0} \end{array}\right\}$$

From Equation (Equation 38), when ξ→0, all the corresponding stress modes that have $\lambda_{i}\geq 1$ will disappear. When $\lambda_{i}=0.5$, singular stresses are obtained in mode I and mode II. An analytical solution of the limits in Equation (Equation 38) yields
(39)$$\left\{\begin{array}{c} K_{I}\\ K_{II} \end{array}\right\}=\sqrt{2\pi L_{0}}\sum_{i=I,II}\left(c_{i}{\left\{\begin{array}{c} \xi^{-\lambda i-1}\sigma_{yy}{|}_{\theta =0}\\ \xi^{-\lambda i-1}\sigma_{xy}{|}_{\theta =0} \end{array}\right\}}_{i}\right)$$

### 2.4. Crack Growth Criterion

The zero-K condition based on [52] is used to determine crack propagation in the crack domain. Therefore, when the stress at the crack tip is finite, a cohesive crack propagates, and accordingly, no stress singularity exists. The crack will propagate in the condition
(40)$$K_{I}\left(\theta \right)\geq 0$$

The crack length Δa and its angle θ are used to define the new location of the crack tip. Figure 2a displays the discretised SBFEM polygon and cracked subdomains of the cohesive crack model (CCM) after the first round of growth. In this shadow domain concept, the crack surfaces is discretized first, and then crack cell elements (CIEs) are inserted into the mesh. This will partition the crack subdomain S1 into two (S1 and S2 in Figure 2b). The CIEs are then used to calculate side-face traction along the crack, upon which the SIFs $K_{I}\left(\theta \right)$ can be defined to calculate the crack growth criterion. We apply the mesh mapping technique to calculate the nodal displacements of the cracked subdomain S1. The remeshing procedure during crack propagation is performed based on [40].

## 3. Cumulative Damage-Plasticity Based Constitutive Law

The constitutive behavior describing cyclic damage in the process zone is embedded in the definition of the interface elements. It has been defined using the thermodynamics-based uniaxial interface model proposed in [46,53,54]. The model assumes that the development of cyclic load is dominated by a cumulative level of the inelastic relative displacement within the interface. The uniaxial model can be applied for the normal behavior ($\sigma_{N}-w$) and for the shear behavior (τ−s) of the interface as a unified constitutive model.

### 3.1. Brief Summary of the Model’s Formulation

The regular formulation of the thermodynamically interface model is described briefly in this section. The Helmholtz free energy is defined as
(41)$$\rho \psi \left(u,u^{P},\omega ,\alpha ,z\right)=\frac{1}{2}\left(1-\omega \right)E{\left(u-u^{P}\right)}^{2}+\frac{1}{2}\gamma \alpha^{2}+\frac{1}{2}Kz^{2}$$
where ρ is the density; *E* is the elastic stiffness; *u* represents the relative displacement at the interface (i.e., opening displacement u=w in the normal direction and slip u=s in the tangential/shear direction); *K* and γ represent the isotropic and kinematic hardening moduli, respectively. The state variables of the interface model are the inelastic displacement $u^{P}$, the damage variable ω, and the hardening variables z,α.

The thermodynamic forces, *X* and *Z*, and the related energy release rate, *Y*, can be calculated by differentiating Equation (Equation 41) with respect to each state variable as follows.
(42)$$\sigma^{P}=\sigma =-\frac{\partial \rho \psi }{\partial u^{P}}=\left(1-\omega \right)E\left(u-u^{P}\right)$$
(43)$$X=\frac{\partial \rho \psi }{\partial \alpha }=\gamma \alpha ,\;\;\;Z=\frac{\partial \rho \psi }{\partial z}=Kz$$
(44)$$Y=-\frac{\partial \rho \psi }{\partial \omega }=\frac{1}{2}E{\left(u-u^{P}\right)}^{2}$$
where σ represents the stress components (i.e., normal stress $\sigma_{N}$ in the case of opening displacement *w*, and shear stress τ in the case of slip displacement *s*). A yield function similar to plasticity theory, which defines the boundary between elastic and inelastic domains, is introduced into the effective stress space as follows.
(45)$$f\left(\tilde{\sigma },X,Z\right)=\left|\tilde{\sigma }-X\right|-Z-\sigma_{0}$$
with $\tilde{\sigma }$ being the effective stress given as $\tilde{\sigma }=\sigma /\left(1-\omega \right)$ and $\sigma_{0}$ being the elastic stress limit. The flow potential determining the damage evolution augments the threshold function (Equation (Equation 45)) with an extra term as
(46)$$\phi =f\left(\tilde{\sigma },X,Z\right)+\frac{S{\left(1-\omega \right)}^{c}}{\left(r+1\right)}{\left(\frac{Y}{S}\right)}^{r+1}$$
where *S* is the damage strength parameter, and *c* and *r* are exponential rate parameters. The evolution equations can be obtained by differentiating (Equation (Equation 46))
(47)$${\dot{u}}^{P}=\dot{\lambda }\frac{\partial \phi }{\partial \sigma^{P}}=\frac{\dot{\lambda }}{1-\omega }\;\mathrm{sign}\left(\tilde{\sigma }-X\right)$$
(48)$$\dot{\alpha }=-\dot{\lambda }\frac{\partial \phi }{\partial X}=\dot{\lambda }\;\mathrm{sign}\left(\tilde{\sigma }-X\right),\;\dot{z}=-\dot{\lambda }\frac{\partial \phi }{\partial Z}=\dot{\lambda }$$
(49)$$\dot{\omega }=\dot{\lambda }\frac{\partial \phi }{\partial Y}=\dot{\lambda }\;{\left(1-\omega \right)}^{c}{\left(\frac{Y}{S}\right)}^{r}$$

This model can be implemented as a time-stepping algorithm, as described in [46]. The damage accumulation under both monotonic and cyclic loading is described through the modified flow potential by [46,54].

### 3.2. Elementary Studies of the Cohesive Model

To illustrate the phenomenological behavior of the used constitutive model and its applicability for modeling cyclic and fatigue behavior, a material model of crack behavior at the point level (Gauss point) under opening and shear displacement is presented in this section.

The described parameters of monotonic and cyclic response material behavior are plotted in Figure 3. The exponential parameter *c* was used to control the dropped-down part of the crack opening (COD) and sliding (CSD) curve. The parameters *c* and *r* were applied for tuning the accumulation of the damage due to cyclic loading. The damage strength parameter *S*, however, could control the brittleness in the response. The model parameters for a common combination of concrete matrix C30/37 were identified using the parametric study reported in [46]. The setup of the study is provided in Figure 3 for monotonic loading and for cyclic loading, as the cohesive model parameters are summarized. The cohesive model stiffness (E) was set equal to Young’s modulus of concrete. The parameters $\bar{\sigma }$, *K*, γ, *S*, *r*, and *c* were identified using a black line for the monotonic response and a blue line for the monotonic response.

The cyclic loading curves of the crack opening versus cyclic loading can be compared with the corresponding curves obtained numerically for monotonic loading. The described model was implemented using zero-thickness interface elements inside the SBFEM framework in Equation (Equation 18)). For the monotonic and the cyclic loading, the damage evolution for the loaded and unloaded responses is depicted in Figure 3 for crack opening and crack sliding. The accumulation of the damage parameter is nonlinear. The traction opening/sliding cohesive models for two loading scenarios are studied.

## 4. Numerical Validation

### 4.1. Test Setup

Three-point bending (TPB) tests were studied to validate the numerical method in this study. The contributions of both traction modes, $k_{n}$ and $k_{s}$, in the cohesive zone model, were investigated. The investigations performed by [8] have shown that the inclusion of the normal energy dissipation dominated the response of post-peak crack mouth sliding displacement (CMOD). The nonlinear equilibrium equations were solved using Newton–Raphson iteration [55], which is characterized by strain softening in the process zone. The benchmark examples are TPB tests with a single-edge notch (Figure 4). Two sizes of the beam were considered in the tests: small beams with a cross-section height of h=200 mm, and large beams with h=400 mm. The beam width was b=100 mm. The lengths of small and large beams were 600 and 1200 mm, respectively. For the notch depth, $h_{0}$ = h/6, whereas the maximum grain size ($d_{0}$) was 8 mm. The experimental measurements for the concrete beams were provided by Baktheer and Becks [8], and the material properties were adopted from [8], as listed in Table 1.

### 4.2. Loading Scenarios

Experimentally, the crack opening displacements and the mid span deflection of the tested beam were recorded, along with the applied force *F*. The TPB supported beam was tested symmetrically by displacement-controlled loading at the top edge. The typical two different loading scenarios are shown in Figure 5. In the SBFEM simulation, an incrementally increased monotonic load (Figure 5a) was applied with an increment size of 0.0005; there were 200 load steps. The load was controlled by the crack tip opening displacement (CMOD) until failure. In the second loading scenario, Figure 5b, a sequence of loading and unloading cycles was applied to define the CMOD. In this way, detailed characteristics of the post-peak loading and unloading of the load–CMOD curve were obtained. This can help to analyze the damage mechanism involved in the cyclic flexural behavior of concrete.

### 4.3. Monotonic Loading

The softening curve parameters to model the fracture process zone are presented, and a range of values were applied based on the parametric study in Section 3. The material parameters were calibrated for two examples under monotonic loading. Then, the material model was validated using the size-effect calculations. The obtained numerical results for cyclic loading were obtained in additional to validating the method. For this investigation, the properties of the concrete and cohesive interface element are listed in Table 2.

The tracked points for the notched pattern and the initial mesh were defined as shown in Figure 4b. For the small beam of cross-section height of h=200 mm, the mesh consisted of 481 polygons and 584 nodes. Meanwhile, for the large beam (h=400 mm), the initial mesh comprised 1483 polygons and 1628 nodes. Plane stress conditions were assumed.

Figure 6 compares the predicted load-crack mouth opening displacement (CMOD) of the TBP small beam with the experimental results reported by [8] under monotonic loading. The corresponding curve of the numerical predictions by SBFEM is depicted in Figure 6, plotted as a blue dashed line. The numerical results of the load–CMOD curve are in a good agreement with the experimental measurements. A maximum load of 18.75 kN was obtained at a CMOD of 0.017 mm. Interestingly, the load–CMOD curve of the numerical was not influenced by the length of crack propagation.

The crack propagation due to increasing load with initial $\Delta_{a}$ = 3.0 mm is shown in Figure 7. Our results show a straight crack path in the direction of the point of external load (F). Th fracture process zone extends up in the middle of the beam Figure 7b at peak load before cracking. At a load of 5.763 kN, the crack propagates in the post-peak region (Figure 7c). For this load level, the cohesive force vanishes. Finally, as the actual crack’s length is increased, the fracture zone is shortened, as expected, by increasing the load level; see Figure 7d. The influence of the size of stiffness degradation is depicted for both small and large tests in Figure 8, which shows the numerical predictions, along with experimental measurements of monotonic tests based on [8]. The nominal strength ($\sigma_{N}$) of SBFEM numerical results were determined in the same way in [8] under monotonic behavior. It is calculated by [1,52]: (50)$$\sigma_{N}=\frac{c_{n}F_{u}}{bh},$$
where $F_{u}$ is the ultimate force and $c_{n}=3L_{0}/\left(2h-h_{0}\right)$ is determined by the bending theory for notched beams. Figure 8 depicts a log–log plot of the the relative size of the beam (horizontal axis) and the nominal strength (vertical axis). The numerical results and the experiments of [8] indicate that the nominal strength is increased by decreasing beam size. The numerical results of nominal strength and the experimental data have a ratio of 1.01–1.04 for small beams, and a ratio of 0.98–1.02 for large beams. In addition, less scatter in the predictions of the large beams was obtained.

### 4.4. Cyclic Loading

Figure 5 shows the numerical predictions and the experimental measurements for cyclically increasing loading. The loading was controlled by the CMOD for three unloading cycles and applied until failure. Good agreement of the numerical predictions (Figure 9b) with respect to the experiment tests (Figure 9a) is obvious.

Furthermore, in our analysis we explore the main dissipative mechanisms. For this purpose, the TPB beams were subjected to a few loading cycles with an incremental increase in the CMOD values. The obtained cyclic responses for both small and large beams are plotted in Figure 10a,b, respectively. One of the principal noticeable effects during the cyclic loading in the post-peak regime is the degradation of the unloading stiffness, which defines the value of the damage. From the damage evolution, it was observed that the damage parameter had a value larger than 0.5 at the first post-peak cyclic load for a small beam; the damage started to progress in the pre-peak subcritical load levels. Furthermore, the damage parameter ω has a value larger than 0.65 for a large beam.

This is explained by knowing that the developed crack showed a rough surface that is not fully closed during the unloading of the specimen. This was confirmed for the cyclic behavior in the simulation of SBFEM and experiment tests. Additionally, the stiffness degradation and the growth of unclosed crack openings were characterized for both sizes.

Plots of $K_{I}$-CMOD are shown in Figure 11 for monotonic and cyclic applied loads. In Figure 11a, the points that represent the initial mesh of Figure 4 were calculated once $K_{I}\geq 0$. Then, the crack opened gradually based on a crack-propagation criterion. The numerical calculation of $K_{I}$ by SBFEM with a fewer degrees of freedom (DOFs) manifested good crack trajectory predictions.

Since the goal of the present study was to apply the constitutive law with a cumulative damage feature within SBFEM, we considered only mode-I cyclic crack propagation in our analysis. Further studies with applications to mixed modes loading are planned for future publications, where more advanced constitutive cohesive zone models could be used, e.g., [14,56].

## 5. Conclusions

Cracks in concrete can occur when the tensile stresses imposed by actions exceed the tensile strength of the material. Furthermore, the cracks can also be initiated under repeated loads with stress levels below the tensile strength. In this work, the cyclic cohesive crack procedure-based SBFEM was implemented to study the crack propagation in concrete. The proposed model showed the ability to simulate the monotonic and cyclic behavior of a cohesive crack interface element, e.g., a concrete interface. It provided a realistic prediction of cyclic damage behavior for up to several load cycles. The output for the numerical simulation of monotonic loading analysis showed full agreement with experimental data from the literature. The results differed 5% for the maximum peak force. Regarding the nominal strength, the ratio of the numerical results to the experimental data under monotonic loading varied between 1.01 and 1.04 for small beams. The ratio was 0.98–1.02 for large beams.

Additionally, the proposed procedure has been proved to be an efficient tool for estimating the damage level. The level of damage accumulation (ω) and material plasticity variables were calculated based on thermodynamics. The described damage model has been successfully implemented to describe the cyclic behavior of cohesive interface elements using SBFEM. The damage parameter ω has a value larger than 0.5 at the first post-peak cyclic load for a small beam, and has a value larger than 0.65 for a large beam. The cyclic responses obtained by SBFEM for both small and large beams presented good agreement with the experimental data.

The predicted load–CMOD responses in the validation examples were within the range measured in the cyclic and monotonic loading experiments. Testing results demonstrated that the most important factors for the overall simulation were the thermodynamic hardening modulus γ and the damage strength parameter *S*. The simulations executed to study the effect of the loading sequence offered successful results and demonstrated the effect of damage accumulation for realistic predictions for concrete structures.

## Figures and Tables

**Figure 1 materials-16-00863-f001:**
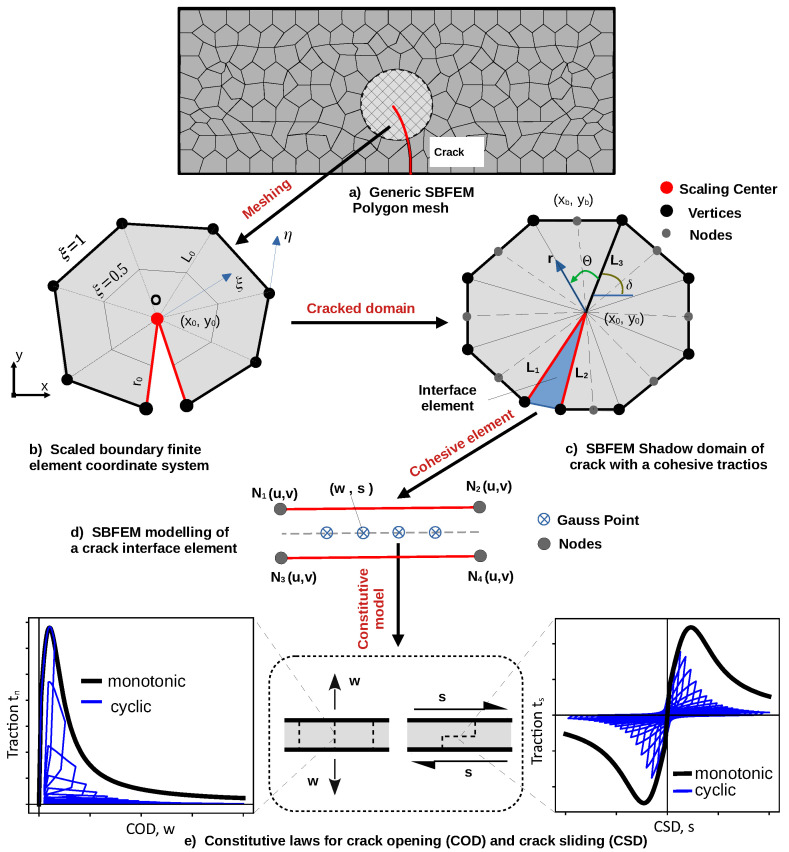
The concept of a cohesive crack model using the scaled boundary finite element method.

**Figure 2 materials-16-00863-f002:**
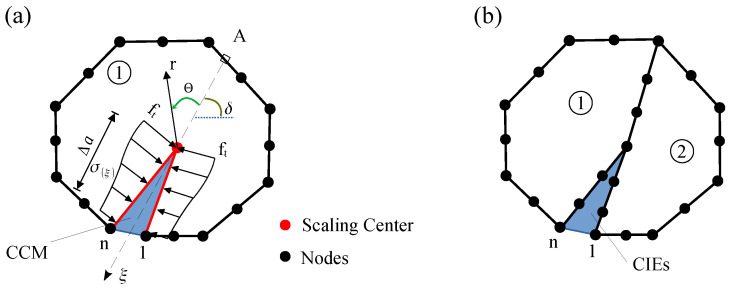
Calculation of $k_{I}$ using shadow domain method: (**a**) cohesive crack model (CCM) in SBFEM; (**b**) subdomain discretization.

**Figure 3 materials-16-00863-f003:**
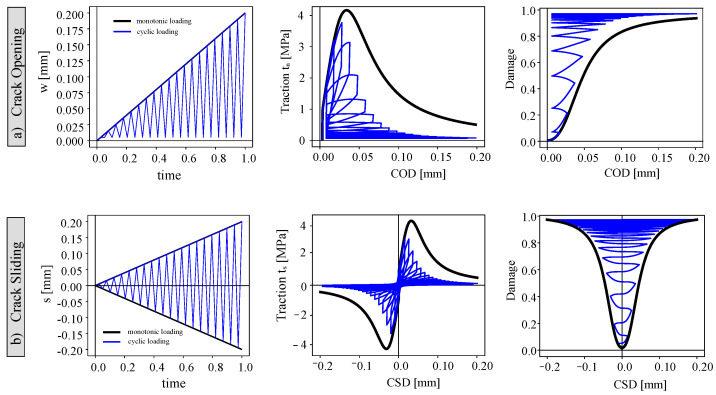
Characterization of the crack behavior under cyclic loading (blue lines) and monotonic loading (black lines) at the material-point level: (**a**) crack opening, (**b**) crack sliding.

**Figure 4 materials-16-00863-f004:**
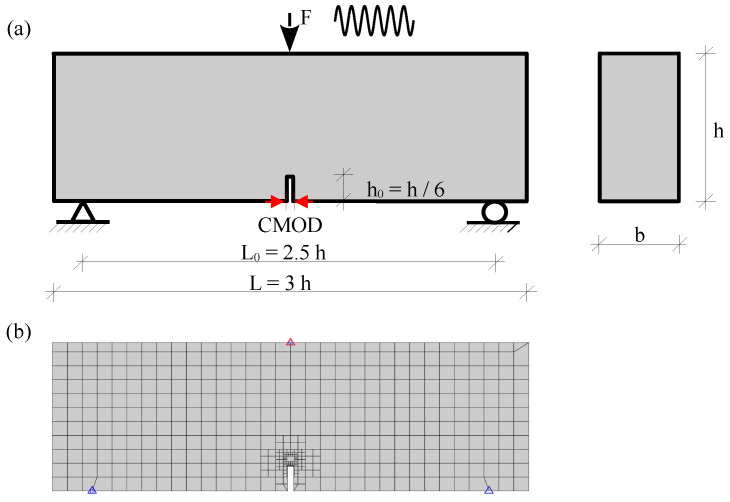
A single-notched concrete beam subjected to a three-point load. (a) Geometry, (b) initial mesh.

**Figure 5 materials-16-00863-f005:**
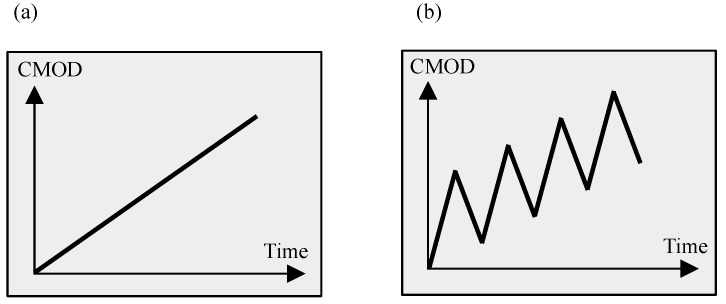
Typical loading scenarios of the studied tested beams: (a) monotonic behavior, (b) cyclic behavior.

**Figure 6 materials-16-00863-f006:**
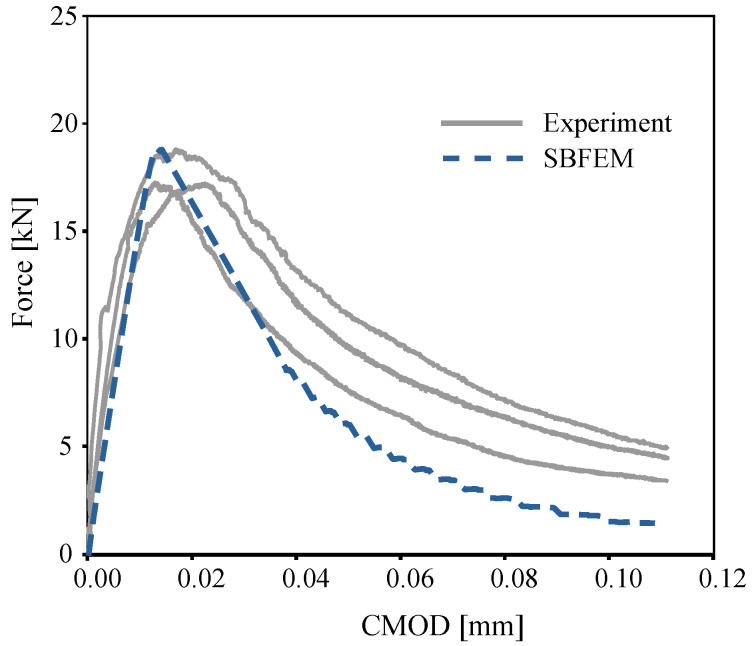
Numerical predictions of load–CMOD curves and the corresponding experimental curves for the single-notched three-point bending test under monotonic loading.

**Figure 7 materials-16-00863-f007:**
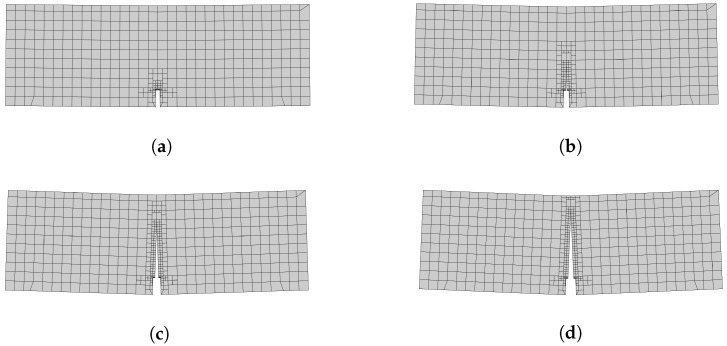
Crack propagation in SBEM subjected to three-point bending tests. (**a**) Load = 8.347 kN (pre-peak), (**b**) load = 18.75 kN (peak load), (**c**) load = 5.763 kN (post-peak), (**d**) load = 2.514 kN (post-peak).

**Figure 8 materials-16-00863-f008:**
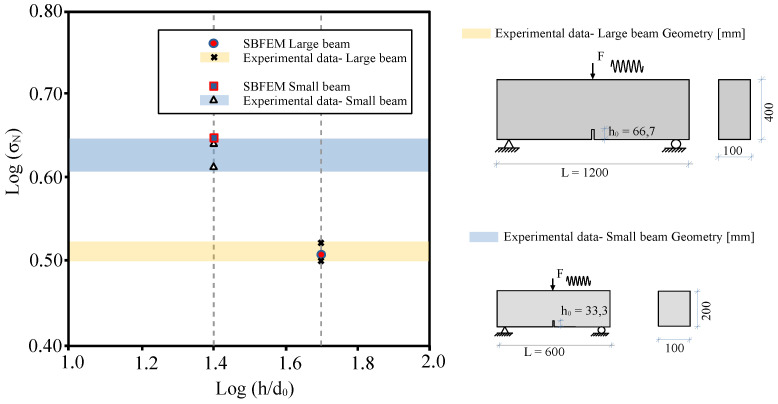
The effect of the size of the beam on the nominal strength under monotonic behavior.

**Figure 9 materials-16-00863-f009:**
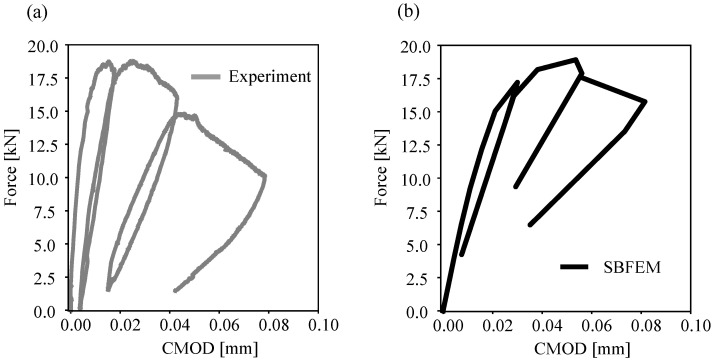
Comparison of numerical predictions (**a**) and experimental measurements (**b**) of Cyclic-CMOD curves for the single-notched three-point bending test.

**Figure 10 materials-16-00863-f010:**
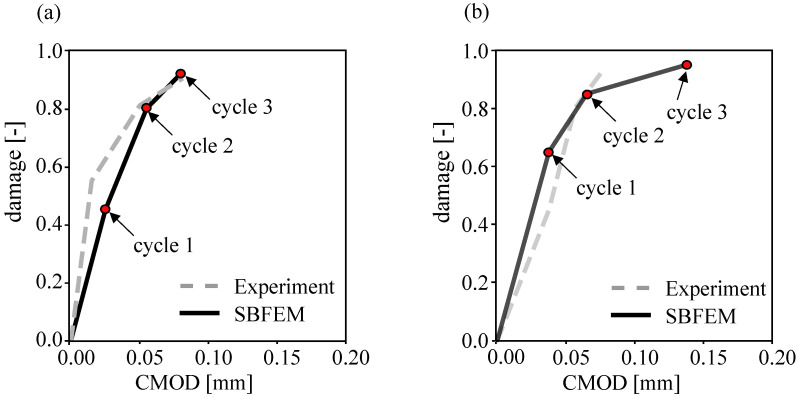
Post-peak cyclic behavior of SBFEM analysis and corresponding damage evolution for (**a**) a small beam and (**b**) a large beam.

**Figure 11 materials-16-00863-f011:**
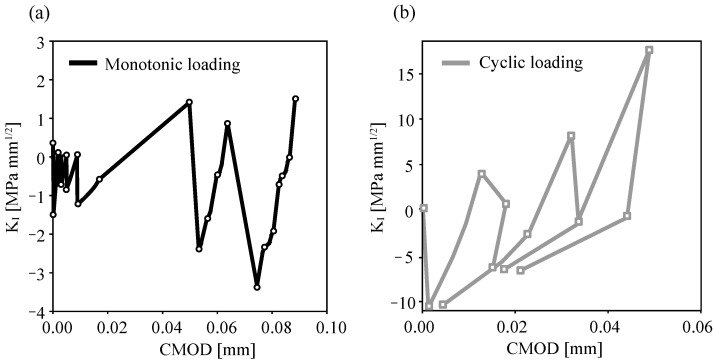
$K_{I}-CMOD$ and loading-point curves for mode-I bending beam for: monotonic loading (**a**) and cyclic loading (**b**).

**Table 1 materials-16-00863-t001:** Parameters of concrete [8].

Parameter	Denomination	Value	Unit
$f_{c}$	Compressive strength	63.61	[MPa]
$f_{ct}$	Tensile strength	4.28	[MPa]
$E_{c}$	Young’s Modulus	34.468	[GPa]
ν	Poisson ratio	0.2	[-]

**Table 2 materials-16-00863-t002:** Model parameters for concrete cohesive interface element.

Parameter	Denomination	Value	Unit
*E*	Elastic cohesive modulus	2800.0	[MPa]
$\bar{\sigma }$	Reversibility limit	1.0	[MPa]
*K*	Isotropic hardening modulus	300.0	[MPa]
γ	Kinematic hardening modulus	200.0	[MPa]
*S*	Damage strength	2.5 × $10^{-4}$	[MPa]
*r*	Damage accumulation parameter	1.0	[-]
*c*	Damage accumulation parameter	0.8	[-]

## Data Availability

Not applicable.

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
