# Peer review of "Modeling Cyclic Crack Propagation in Concrete Using the Scaled Boundary Finite Element Method Coupled with the Cumulative Damage-Plasticity Constitutive Law"

_materials, 2023, doi:10.3390/ma16020863_

Round 1
Reviewer 1 Report
The manuscript has some information for the readers and needs some revision for consideration.
1. Motivation behind the study and the novelty must be highlighted.
2. Why particularly SBFEM is considered for the study. Provide proper explanation.
3. The references are used, without detailed analysis. Constitute a relation between the references and your work.
4. In SBFEM theory, many equations are used without background analysis. Proper references must be provided wherever necessary.
5. More explanation is needed for Fig. 1.
6. How cohesive crack model is superior?
7. How loading conditions are selected for testing?
8. Avoid using figures in the conclusion section.
9. Improve the conclusions by using more numerical values.
Author Response
Dear Editor,
please find attached file our detailed response to the valuable suggestions and
comments. The revised parts and additions in the manuscript text were applied and highlighted.
Sincerely,
Omar Alrayes

Reviewer 2 Report
This manuscript cyclic carack growth were analyzed by BFEM as well as CDPC law in the concrete materials. The title is interesting and the manuscript is well written. Some suggestions and questions are provided below to be applied and answered:
- The final sentences of the abstract can be designated to more quantitative results.
- It is suggested to convert the presented steps in Page 5 to a flowchart.
- The dashed lines around the figures 1, and 2 are not necessary.
- There are some other related publications which can be added to the literature review. Some of the related publications provided below:
· Daneshyar, A., Sotoudeh, P., & Ghaemian, M. (2022). The scaled boundary finite element method for dispersive wave propagation in higher‐order continua. International Journal for Numerical Methods in Engineering.
· Daneshvar, M. H., Saffarian, M., Jahangir, H., & Sarmadi, H. (2022). Damage identification of structural systems by modal strain energy and an optimization-based iterative regularization method. Engineering with Computers, 1-21.
· Jiang, X., Zhong, H., Li, D., & Chai, L. (2022). Dynamic Fracture Modeling of Impact Test Specimens by the Polygon Scaled Boundary Finite Element Method. International Journal of Computational Methods, 2143010.
- The conclusion can be enriched by adding more quantitative outcomes.
Author Response

(The authors gave the same response as above.)

Reviewer 3 Report
The work presents an interesting numerical and experimental study about a numerical modeling approach for simulating crack growth using scaled boundary finite elements. The figures have excellent quality. I detail some doubts to improve the manuscript.
Line 2: Cracks in concrete structures also are from bending. It is interesting to add this fact.
Equation (1): What is the degree of the adopted shape functions?
Line 99: Replace “manuscript” with “work”.
Line 151: Is “very fine mesh” a more discretized mesh?
Line 207: C30/37 concrete matrix. Are the numbers the compressive strengths of concrete?
Line 215: Doesn't using zero-thickness cause singularity? Division by zero. Explain
Line 236: What is the size or percentage of the increments?
Line 254: 1483 polygons is the number of finite elements?
Table 2: Are the parameters from any reference?
Figure 8: Write the beam cross-section dimensions.
Figure 9: Combining the two images in one would more clearly show the agreement between experimental and numerical values.
Author Response

(The authors gave the same response as above.)

Round 2
Reviewer 1 Report
Many of the equations are used and must be checked for correctness before acceptance.
Reviewer 3 Report
The authors have answered my questions and accepted my suggestions, so the paper has been improved. Therefore, it can be accepted for publication.